# Comprehensive Perspective for Lung Cancer Characterisation Based on AI Solutions Using CT Images

**DOI:** 10.3390/jcm10010118

**Published:** 2020-12-31

**Authors:** Tania Pereira, Cláudia Freitas, José Luis Costa, Joana Morgado, Francisco Silva, Eduardo Negrão, Beatriz Flor de Lima, Miguel Correia da Silva, António J. Madureira, Isabel Ramos, Venceslau Hespanhol, António Cunha, Hélder P. Oliveira

**Affiliations:** 1Institute for Systems and Computer Engineering, Technology and Science, INESC TEC, 4200-465 Porto, Portugal; joana.p.morgado@inesctec.pt (J.M.); francisco.c.silva@inesctec.pt (F.S.); acunha@utad.pt (A.C.); helder.f.oliveira@inesctec.pt (H.P.O.); 2Centro Hospitalar e Universitário de São João, CHUSJ, 4200-319 Porto, Portugal; claudiaasfreitas@gmail.com (C.F.); eduardo.negrao@gmail.com (E.N.); beatrizflordelima@hotmail.com (B.F.d.L.); miguel.ncds@gmail.com (M.C.d.S.); antonio.madureira@chsj.min-saude.pt (A.J.M.); iramos@med.up.pt (I.R.); hespanholv@gmail.com (V.H.); 3Faculty of Medicine, University of Porto, FMUP, 4200-319 Porto, Portugal; jcosta@ipatimup.pt; 4Institute for Research and Innovation in Health of the University of Porto, i3S, 4200-135 Porto, Portugal; 5Institute of Molecular Pathology and Immunology of the University of Porto, IPATIMUP, 4200-135 Porto, Portugal; 6Faculty of Science, University of Porto, FCUP, 4169-007 Porto, Portugal; 7Department of Engineering, University of Trás-os-Montes and Alto Douro, UTAD, 5001-801 Vila Real, Portugal

**Keywords:** lung cancer assessment, tumour characterisation, personalised medicine, computer-aided decision, computed tomography analysis

## Abstract

Lung cancer is still the leading cause of cancer death in the world. For this reason, novel approaches for early and more accurate diagnosis are needed. Computer-aided decision (CAD) can be an interesting option for a noninvasive tumour characterisation based on thoracic computed tomography (CT) image analysis. Until now, radiomics have been focused on tumour features analysis, and have not considered the information on other lung structures that can have relevant features for tumour genotype classification, especially for epidermal growth factor receptor (*EGFR*), which is the mutation with the most successful targeted therapies. With this perspective paper, we aim to explore a comprehensive analysis of the need to combine the information from tumours with other lung structures for the next generation of CADs, which could create a high impact on targeted therapies and personalised medicine. The forthcoming artificial intelligence (AI)-based approaches for lung cancer assessment should be able to make a holistic analysis, capturing information from pathological processes involved in cancer development. The powerful and interpretable AI models allow us to identify novel biomarkers of cancer development, contributing to new insights about the pathological processes, and making a more accurate diagnosis to help in the treatment plan selection.

## 1. Introduction

Lung cancer is still the leading cause of cancer death in the world as a result of high incidence combined with low 5-year survival rates [1,2]. For these reasons, lung cancer deserves special attention from the medicine, biology, and scientific communities in order to develop novel solutions to increase the early diagnosis, assist in treatment decisions, and monitor responses to improve patient outcomes. The molecular profile of the tumour tissues enables the identification of driver mutations, and targeted therapies can be used for particular genotypes. Traditional chemotherapy works by killing all cells, without discriminating between normal and cancerous cells. Instead, targeted therapy acts in specific elements, interfering with the cancer driver genes and stopping or slowing the growth of tumour cells.

Epidermal growth factor receptors (EGFRs) and Kirsten rat sarcoma viral oncogenes (KRASs) are the most frequently mutated genes present in non-small-cell lung cancer (NSCLC) [3,4,5], which is a major sub-type of lung cancer [6]. Activating mutations in EGFRs (namely exon 19 deletions or exon 21 L858R point mutations) benefit from treatment with EGFR tyrosine kinase inhibitors (TKIs) [7,8,9]. This gene is responsible for multiple biological processes and is useful to determine the clinical outcomes in many lung diseases. Abnormalities in EGFR pathways cause abnormal EGFR signalling and are associated with cancer, lung fibrosis, and numerous airway diseases [10]. Targeted therapies have been studied in recent years, with encouraging results for EGFRs [11,12], improving progression-free survival for patients with advanced NSCLC who were selected on the basis of EGFR mutations [12,13,14,15]. EGFR-dedicated therapies are currently used as first- and second-line lung cancer treatments [16], and several others are in development [17]. On the other hand, mutant KRAS has a wide spectrum of other co-occurring genetic alterations and a high biological heterogeneity, including diverse KRAS point mutations, which hinder the development of new target therapies [18]. For mutated KRAS, there are no current clinically approved targeted therapies, but there are several KRAS inhibitors in clinical trials [19,20,21]. Additionally, another target therapy of NSCLC has emerged—immunotherapy. This therapy relies on the use of immune checkpoint inhibitors to release the patient’s immune cells to fight the cancer [22]. Although it has demonstrated significant patient improvement, only a small portion of patients benefit from this therapy (20%) [23]. This is attributed to the low performance of the current predictive biomarkers of response to immune checkpoint blockade therapy, which rely on detection of programmed death ligand 1 (PD-L1) in cancer tissue [24]. Tumour-infiltrating immune cells are a key population of the tumour microenvironment and mediate the antitumor effects of immunotherapy [25]. The classification of the different immune cells helps to better define the immunogenic potency of NSCLC [26]. Despite the evident benefits, with the increased use of these personalised therapies in oncology, new side effects have emerged, causing important clinical challenges in the management of lung cancer patients. In fact, although the majority of these events are mild, some of them can be severe and potentially life-threatening [27].

Tissue biopsy is the traditional method to identify the main biomarkers of the tumour [28]; however, it is an invasive procedure with clinical implications such as pneumothorax, pain, and complications like infection, haemorrhage, and damage to surrounding tissues [29]. Due to the importance of tumour characterisation, less invasive, easier, and faster techniques to access the genotype of the tumour are needed. Computed tomography (CT) plays a key role in lung cancer management from initial diagnosis and staging to treatment response assessment [30]. CT is more sensitive than chest radiography in lung cancer screening [31,32,33]. Moreover, it allows for a three-dimensional (3D) thorax characterisation as each nodule is assessed, and information about other lung structures can be retrieved. The application of artificial intelligence (AI) solutions for lung cancer imaging has been dedicated to reproducing the radiology procedure. The traditional computer-aided decision (CAD) support system approaches based on a CT scan start with nodule detection and segmentation for further analysis, e.g., malignancy and subtype classification [32,34,35]. AI-based solutions dedicated to predicting the risk of LCa showed high-performance results and represent an opportunity to optimise the screening process, reducing the false positives and false negatives on assessments performed by radiologists [35]. On the genotype characterisation, previous studies using the radiomic features from CT images have shown that it is possible to use the imaging information to predict the gene mutation status related to cancer development [36,37,38,39,40]. However, the majority of these studies are focused on the nodule, which cannot capture the extension and complexity of the pathophysiological phenomena that occur in the other structures, but that can be related to the cancer development—and could introduce insights useful for the diagnosis and prognosis of the patient.

The general idea for AI-based solutions is to follow the same procedure but improve the accuracy of the diagnosis by trying to detect missed nodules, reducing the time and effort of clinicians during the evaluation process, and creating a measurable impact on clinical management and patient outcomes. Furthermore, AI-based solutions enable the identification of radiomic information from the lung structures on the CT images that are not visible to the naked eyes of radiologists, producing a more accurate genotype characterisation. In fact, this comprehensive perspective has not yet been explored for CADs in lung cancer. Recent works studied other lung structure abnormalities and found relevant relations with the presence of lung cancer. The current work identifies the studies of the most important pathophysiological processes in the lung related to lung cancer and opens the discussion about what kind of information can be decisive for these novel and comprehensive radiomic approaches. The next generation of CADs aim to integrate more representative information about the relevant biological changes in the lung to produce predictive models that can improve the accuracy of the tumour characterisation (avoiding the need for biopsy) in order to help therapeutic decisions and leverage personalised medicine—selecting patients who will definitely benefit from targeted therapies and avoiding superfluous therapy-related side effects.

## 2. Pathophysiologic Features

Genotype characterisation, as an essential step for treatment decision, may benefit if more information about the simultaneous pathophysiological processes that occur is combined with traditionally used nodule information. Recent studies have identified other relevant biological structures beyond the nodule that can help the tumour characterisation and contribute to a better understanding of cancer development [41,42]. In fact, lung cancer has been studied as a more extensive clinicopathological phenomenon that involves several other lung structure alterations, and their relationship with cancer development has been identified. The main pathophysiological changes related to lung cancer can be identified on specific findings in the CT images. The most relevant examples are represented in Figure 1: emphysema, pulmonary fibrosis, air bronchogram, pleural retraction, and vascular convergence.

Emphysema causes damage to the alveoli, and, as a consequence, there is a reduction in the gas exchange efficiency [43]. On CT images, emphysema is characterised by a compartment of air seen at extremely low attenuation areas (Figure 1a) [44]. Chronic inflammation in the airways has been shown to be important to the pathogenesis of both emphysema and lung cancer [45,46,47,48,49], and it is recommended to consider emphysema when assessing lung cancer risk [45,46].

Pulmonary fibrosis affects the tissue surrounding the alveoli (interstitium), and this condition occurs when lung tissue becomes thick and stiff [50]. The three specific findings for fibrosis are: traction bronchiectasis, loss of volume, and honeycomb (Figure 1b) [51]. Fibrosis might contribute to carcinogenesis due to the occurrence of atypical or dysplastic epithelial changes that progress to invasive malignancy [52].

Air bronchogram is characterised by a pattern of air-filled bronchi on the background of a nodular opacity [53]. The airways appear in the CT images as air-filled structures that originate an opacification of the surrounding alveoli (Figure 1c) [54]. The correlation between air bronchogram and lung cancer has been studied, and CT air bronchogram is an important malignant feature to predict the invasiveness of lung cancer [55,56].

Pleural retraction consists of pulling the visceral pleura toward the invading neoplastic tissue [57], and can be identified in the CT images as millimetre-thin lines of spun pleura (Figure 1d). Pleural retraction is correlated with lung cancer [58], and with the EGFR mutation [59,60].

Vascular convergence is verified when the vessels converge to a nodule without adjoining or contacting the edge of the nodule (Figure 1e) [57,61,62]. This phenomenon reflects angiogenesis [57].

Angiogenesis is essential for tumour growth and metastasis; maybe, for this reason, the convergence of vasculature towards or surrounding a nodule is related to lung cancer stage and pathology [61].

## 3. Comprehensive Perspective for the Next Generation of CADs

The correlation between several pathophysiological changes in the lung has been studied, since those phenomena do not occur in isolation and usually share pathways and/or functional mechanisms with lung cancer development. EGFR regulates several biological processes, and the correlations between different lung pathologies have been suggested. This is an important point and can expand the region of interest (ROI) for predictive models.

Radiomics in lung cancer have mostly been based on nodule assessment [32,63]. Recently, a few approaches have tried to use features from other lung structures; however, this information came from semantic annotations provided by radiologists [41,42,59]. As AI has shown to be able to extract and use relevant information, the quantitative analysis of the lung cancer performed by radiomic analysis could improve the performance on tumour characterisation—if the data from relevant structures of the lung were taken into consideration and selected by automatic feature learning methods, avoiding the human effort required for semantic annotations and making the process less subjective. The next generation of CADs should be able to use large lung regions for analysis and feature learning in order to capture more information associated with cancer pathogenesis. The most powerful learning methods are based on deep learning techniques, which allow for the capture of information that is not visible to the naked eye and avoid ad hoc feature extractions, depending on the feature engineering processes used for the task. Furthermore, those methods allow us to cope with the wide heterogeneities of clinical data using massive databases. Based on these advantages, the AI-based models for novel and comprehensive CADs would be mainly based on deep learning algorithms. Additionally, explainable AI, based on activation maps, can identify which part in the medical image was used to contribute to the final classification from the learning model [64,65,66]. AI solutions will move from “black boxes” to interpretable models that will help clinicians to understand which are the regions and features that contribute to the final decision, build trust in the methods to use in the clinical context, and create a deeper understanding of pathological features that will facilitate the management of lung cancer [67].

Figure 2 represents the main differences between the traditional CADs and the novel and comprehensive approach of next-generation CADs. The current solutions for automatic imaging analysis are: focus on the nodule assessment, mainly for screening; and the need of a biopsy for the tumour characterisation. Novel CADs should be able to detect the malignancy and characterise the lung tumour based on CT images and clinical data, using explainable models that help clinicians to understand the choice made by the model. However, despite not being represented in Figure 2, even the new AI-based solutions will still need to use biopsy as a backup method for cases in which radiomics would not be conclusive on tumour characterisation.

There are three main levels of actions for CADs: screening, diagnosis/characterisation, and treatment assignment. On treatment planning, the comprehensive approach would assist clinicians in choosing the optimal treatment, with clear and transparent explanations for recommendations. Currently, recommendations are clear about the need for biopsy to determine the mutational status of the tumour in order to select the best available treatment [68]. However, learning from a comprehensive approach offers the possibility of capturing correspondences between processes and gaining an in-depth understanding of pathophysiological changes. This would allow for the selection of a personalised treatment that will improve effectiveness and efficiency while diminishing avoidable therapy-related adverse events. The AI-based approach would stratify the patient groups according to their singular properties in order to more effectively evaluate the potential efficacy of treatments, recommend the sequence of therapies, and predict the effects of specific drugs and clinical outcomes. This strategy may be particularly helpful in elderly or unfit patients who are at higher risk of procedure-related complications. In these patients, such risks may interfere with a proper diagnosis or treatment. For example: since inoperable early stage lung cancer patients may benefit from stereotactic radiotherapy [69], a noninvasive diagnosis strategy in this group of patients could diminish the risks, prevent complications associated with traditional methods, and allow for prompt treatment, thus leading to better disease control and survival rates. Moreover, CADs would use aggregated knowledge of many patients with matching results, history, biomarkers, physiological characteristics, and behavioural risk to present clinicians with the most efficacious treatment option. In addition to preventing toxicities, personalised treatment recommendations can reduce time loss and costs.

The two approaches represented in Figure 2 have pros and cons based on the specific features of those approaches and the stage of development (Table 1). The traditional lung cancer characterisation (biopsy) is an invasive procedure that limits repetition for treatment response evaluation, and automatic solutions are not currently significantly used in the screening process. The comprehensive lung cancer characterisation using imagiological data represents a noninvasive option, avoiding all the complications associated with an invasive procedure. For this reason, it can be repeated several times in order to assess the treatment response and personalise a treatment plan. The novel approaches are in development with the emergence of powerful AI-based models that can give interpretable information that can be helpful when identifying relations between pathophysiological processes that occur during cancer development. Novel radiomic approaches will allow for the identification of the main biomarkers and will study the relationship between the imagiological findings and the lung cancer development, creating a comprehensive analysis of pathophysiological processes.

The biggest challenge for these comprehensive models, which will use information from multiple lung structures, comes from the size of the dataset used for training. Due to the high degree of variability that can be found in lung structures, there is a need for a massive amount of data that covers all heterogeneities, in order to create a good representation of the population [70]. Only with large datasets will it be possible to capture the relevant information, correlate pathophysiological phenomena between structures, and create a better understanding of the processes and mechanisms involved in cancer development. The analysis based on multiple structures has not yet proven conclusive, but certainly deserves additional studies since multiple pathophysiological processes share common pathways and have been shown to be related [47].

The need for large datasets is a transverse limitation on AI-based solutions in healthcare. The ImageNet, composed of 14 million natural images, allowed for the training of complex and powerful neural networks and consequently revolutionised image classification [71]. Currently, the biomedical field is struggling with data size limitations and attempting to build robust models to help clinicians with diagnoses. Large medical datasets are extremely difficult to obtain due to privacy and security issues, annotation efforts by experts, and the huge investment required to collect, store, and maintain the data. The reuse of clinical data in data banks will allow for an important improvement in deep learning solutions for healthcare. The LIDC Data Collection Process for Nodule Detection and Annotation was used in multiple publications [72], allowing for the development of the most relevant radiomic studies for lung cancer screening using CT images [63]. This shows the importance of large datasets in leveraging the development of AI tools. With large datasets, which cover all of the heterogeneities in the population, it will be possible to study the importance of other lung structures for lung cancer characterisation.

## 4. Conclusions

Several previous works showed the relationship between pathophysiological changes in lung structures and lung cancer development, which suggests that there are common biological pathways that can be captured by CT images and used by comprehensive and automatic systems to characterise lung cancer. The dataset size (under-representative of the population) is still the biggest limitation on the development of powerful methods to cope with the heterogeneities of all lung structures. Some recent works have already tried to include more information than the nodule features; however, the small datasets (hundreds of patients) were not representative of all of the variabilities. For this reason, they did not achieve relevant performance improvement. Even so, they confirm the relevance of those other structures. These relations can be used to study the mechanism of cancer development. For radiomic-based solutions, the integration of novel information will allow for the development of a more comprehensive assessment and more accurate models, leading to better tumour characterisation and personalised treatment plans. Understanding the mechanisms that drive cancer processes with other pulmonary diseases—along with better disease models—is essential for the development of new targeted treatments.

## Figures and Tables

**Figure 1 jcm-10-00118-f001:**
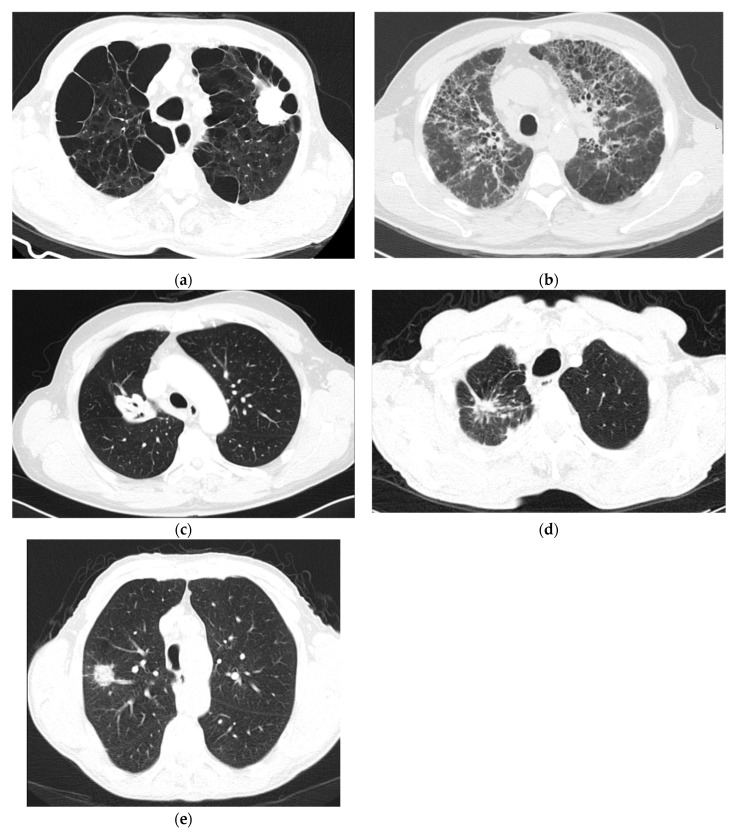
A set of axial computed tomography (CT) images with findings related with lung cancer pathogenesis: (**a**) centrilobular emphysema; (**b**) pulmonary-fibrosis; (**c**) air bronchogram; (**d**) pleural retraction; and, (**e**) vascular convergence.

**Figure 2 jcm-10-00118-f002:**
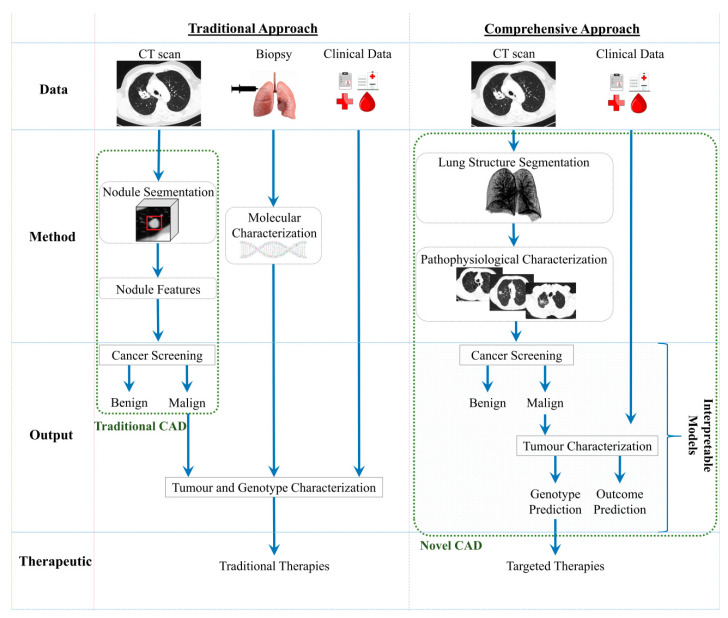
Conceptual diagram of the traditional and comprehensive approaches based on CADs. CT, computed tomography; CAD, computer-aided decision.

**Table 1 jcm-10-00118-t001:** Pros and Cons of traditional and comprehensive approaches for lung cancer diagnosis.

	Traditional Approach	Comprehensive Approach
Pros	-Clinically validated	-Non-invasive assessment, faster and with lower costs; -Safety repeated; -Leverage the personalised medicine; -Interpretable models; -Comprehensive perspective
Cons	-Invasive and with clinical implications; -Restriction for the repetitions of the procedure; -AI based solutions with residual help in the diagnosis	-In development; -Requirement large datasets to train the predictive models

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
