# Peer review of "Comprehensive Perspective for Lung Cancer Characterisation Based on AI Solutions Using CT Images"

_jcm, 2020, doi:10.3390/jcm10010118_

Round 1

Reviewer 1 Report

The authors provide their perspective on computer-aided diagnosis in lung cancer today and in the near future, in particular focusing on the idea of using more than just the nodule information. The authors briefly review the concepts of genetic mutations and targeted immunotherapy, providing a justification for the use of AI for tumor and genotype characterization.

The authors attempt to breakdown the field of lung cancer characterization into "traditional" and "comprehensive" approaches, where the "comprehensive" approaches represent the use of AI that uses information from the entire CT scan to characterize the tumor, leading to the use of targeted therapies. The pros and cons of each approach are described. The authors recognize that the biggest challenge facing the development of AI approaches is the lack of large databases.

This is a good start to the topic, but there are many limitations to the article, especially for a "comprehensive" perspective. The authors do not define what they mean by artificial intelligence -- I think the authors only mean "deep-learning" based AI approaches, but the term AI also includes machine learning algorithms such as neural networks and support vector machines using features. The authors seem to imply that AI is only used in the new "comprehensive" approaches and not in traditional CAD, but many traditional CAD approaches make use of machine learning algorithms.

The authors do not seem to address the body of previous work that has used radiomic features to predict EGFR mutations (a few examples: https://doi.org/10.1002/mp.13747, https://doi.org/10.1016/j.cllc.2016.02.001, https://doi.org/10.1111/1759-7714.13163, https://link.springer.com/article/10.1007/s00330-019-06024-y, https://doi.org/10.1016/j.lungcan.2019.03.025), which leads to the Figure 2 suggesting that only the new "comprehensive" approach is able to provide genotype prediction. This also leads to the authors stating that the "traditional" approach only does molecular characterization using biopsy -- I would suggest that the biopsy be addressed separately, as you could very well do the same thing in the comprehensive Ai approach.

A key article from 2019 on the use of deep-learning in lung cancer screening also is not mentioned in this article: https://www.nature.com/articles/s41591-019-0447-x/. While the focus of that article is on predicting lung cancer and not necessarily on the phenotype, it should be mentioned in a comprehensive article on AI.

There are also a few awkward sentences throughout the article. Line 22, I'd switch "allows" to "enables".  Line 24, "discriminate" should be "discriminating". Line 66-68, "…AI-based solutions could allow to identify the radiomic information.." could be better worded. Lines 136-137, "… even for the new AI-based solutions will persist the need…" could be reworded to ".. Even the new AI-based solutions will still need.."

Author Response

Comments and Suggestions for Authors

The authors provide their perspective on computer-aided diagnosis in lung cancer today and in the near future, in particular focusing on the idea of using more than just the nodule information. The authors briefly review the concepts of genetic mutations and targeted immunotherapy, providing a justification for the use of AI for tumor and genotype characterization.

The authors attempt to breakdown the field of lung cancer characterization into "traditional" and "comprehensive" approaches, where the "comprehensive" approaches represent the use of AI that uses information from the entire CT scan to characterize the tumor, leading to the use of targeted therapies. The pros and cons of each approach are described. The authors recognize that the biggest challenge facing the development of AI approaches is the lack of large databases.

Author response: We thank the reviewer for the positive feedback that we greatly appreciate.

Reviewer#1, Concern # 1: This is a good start to the topic, but there are many limitations to the article, especially for a "comprehensive" perspective. The authors do not define what they mean by artificial intelligence -- I think the authors only mean "deep-learning" based AI approaches, but the term AI also includes machine learning algorithms such as neural networks and support vector machines using features. The authors seem to imply that AI is only used in the new "comprehensive" approaches and not in traditional CAD, but many traditional CAD approaches make use of machine learning algorithms.

Author response: We thank the reviewer for this insight point about the AI term. We agree that many traditional CADs are based on machine learning, and we mention several of them in this new version of the manuscript. However, for the novel CADs, the AI approaches are mainly based on deep learning models since those are the methods able to capture features that are normally invisible to the naked eye, which enables to avoid the ad hoc feature extraction. Additionally, those methods allow coping with the wide heterogeneities of data using massive databases. The authors added the following sentences to the manuscript in order to clarify.

“The most powerful learning methods are based on deep learning techniques, which allow to capture information that cannot be visible to the naked eyes and avoid ad hoc feature extraction dependent on the feature engineering processes used for the task. Furthermore, those methods allow coping with the wide heterogeneities of clinical data using massive databases. Based on these main advantages, the AI-based models for novel and comprehensive CADs would be mainly based on deep learning algorithms.”

Reviewer#1, Concern # 2: The authors do not seem to address the body of previous work that has used radiomic features to predict EGFR mutations (a few examples: https://doi.org/10.1002/mp.13747, https://doi.org/10.1016/j.cllc.2016.02.001, https://doi.org/10.1111/1759-7714.13163, https://link.springer.com/article/10.1007/s00330-019-06024-y, https://doi.org/10.1016/j.lungcan.2019.03.025), which leads to the Figure 2 suggesting that only the new "comprehensive" approach is able to provide genotype prediction. This also leads to the authors stating that the "traditional" approach only does molecular characterization using biopsy -- I would suggest that the biopsy be addressed separately, as you could very well do the same thing in the comprehensive Ai approach.

Author response: We thank the reviewer for this suggestion. The suggested references were added to the manuscript since they represent works relevant to the subject matter, which were missing from the original manuscript. The traditional approaches are mainly focused on nodule, which can be used for oncogene mutation status classification but cannot provide a comprehensive analysis of the pathophysiological phenomena associated with lung cancer development, which already showed to be extensive and related to other lung structures. This work explores the relations already identified between lung risk and pathophysiological changes. This comprehensive analysis would allow to build more robust methods to identify the biomarkers like the oncogene mutation status or PD-L1. We added the following information to the Introduction section:

“On the genotype characterization, previous studies using the radiomic features from CT images have shown that it is possible to use the imaging information to predict the gene mutation status related to cancer development [1]–[5]. However, the majority of these studies are focused on the nodule, which cannot capture the extension and complexity of the pathophysiological phenomena that occur in the other structures but, which can be related to the cancer development and could introduce useful insights for the diagnosis and prognostic of the patient.”

Reviewer#1, Concern # 3: A key article from 2019 on the use of deep-learning in lung cancer screening also is not mentioned in this article: https://www.nature.com/articles/s41591-019-0447-x/. While the focus of that article is on predicting lung cancer and not necessarily on the phenotype, it should be mentioned in a comprehensive article on AI.

Author response: We agree with the reviewer on the importance of this article. We added this reference to the manuscript.

“Traditional computer-aided decision support systems based on CT scanning start with the nodule detection and segmentation for further analysis, such as malignancy and sub-type classification [6]–[8]. AI-based solutions dedicated to predict the risk of LCa have shown high-performance results and represent an opportunity to optimize the screening process by reducing the false positives and false negatives in the radiologists’ assessment [8].”

Reviewer#1, Concern # 4: There are also a few awkward sentences throughout the article. Line 22, I'd switch "allows" to "enables". Line 24, "discriminate" should be "discriminating". Line 66-68, "…AI-based solutions could allow to identify the radiomic information.." could be better worded. Lines 136-137, "… even for the new AI-based solutions will persist the need…" could be reworded to ".. Even the new AI-based solutions will still need.."

Author response: We thank the reviewer for identifying those examples that allow us to improve the manuscript. We agreed and changed all the cases suggested by the reviewer.

References

[1]      S. Li, C. Ding, H. Zhang, J. Song, and L. Wu, “Radiomics for the prediction of EGFR mutation subtypes in non-small cell lung cancer,” Med. Phys., 2019.

[2]      Y. Liu et al., “Radiomic Features Are Associated With EGFR Mutation Status in Lung Adenocarcinomas,” Clin. Lung Cancer, 2016.

[3]      X. Wang et al., “Decoding tumor mutation burden and driver mutations in early stage lung adenocarcinoma using CT-based radiomics signature,” Thorac. Cancer, 2019.

[4]      T. Y. Jia et al., “Identifying EGFR mutations in lung adenocarcinoma by noninvasive imaging using radiomics features and random forest modeling,” Eur. Radiol., 2019.

[5]      W. Tu et al., “Radiomics signature: A potential and incremental predictor for EGFR mutation status in NSCLC patients, comparison with CT morphology,” Lung Cancer, 2019.

[6]      B. Al Mohammad, P. C. Brennan, and C. Mello-Thoms, “A review of lung cancer screening and the role of computer-aided detection,” Clinical Radiology. 2017.

[7]      A. El-Baz et al., “Computer-aided diagnosis systems for lung cancer: Challenges and methodologies,” International Journal of Biomedical Imaging. 2013.

[8]      D. Ardila et al., “End-to-end lung cancer screening with three-dimensional deep learning on low-dose chest computed tomography,” Nat. Med., 2019.

Reviewer 2 Report

This paper is a review paper presented Lung Cancer based on AI solutions using CT images. It is written safely and well overall and is particularly conscious of EGFR mutation-positive lung cancer. 

It is true that data has been accumulated for diagnostic imaging of lung cancer, which is an indication for molecular-targeted therapy, but it may be hesitant to administer the drug to patients. It can cause treatment-related deaths and the drugs are expensive. For now, it is considered that a biopsy should be performed in order to actually perform drug therapy. On the other hand, radiotherapy such as stereotactic irradiation of elderly patients is often treated without a definitive pathological diagnosis. It would be helpful for the attending physician if the diagnosis could be made by CAD at such times. As mentioned above, what about the fact that there are still some difficult aspects during drug therapy and that it helps during radiotherapy, in addition to the text and table.

Author Response

Comments and Suggestions for Authors

This paper is a review paper presented Lung Cancer based on AI solutions using CT images. It is written safely and well overall and is particularly conscious of EGFR mutation-positive lung cancer. 

It is true that data has been accumulated for diagnostic imaging of lung cancer, which is an indication for molecular-targeted therapy, but it may be hesitant to administer the drug to patients. It can cause treatment-related deaths and the drugs are expensive. For now, it is considered that a biopsy should be performed in order to actually perform drug therapy. On the other hand, radiotherapy such as stereotactic irradiation of elderly patients is often treated without a definitive pathological diagnosis. It would be helpful for the attending physician if the diagnosis could be made by CAD at such times. As mentioned above, what about the fact that there are still some difficult aspects during drug therapy and that it helps during radiotherapy, in addition to the text and table.

Author response: The authors thank the reviewer for this relevant insight. We have added a deeper discussion of the assistance that can be provided by the new comprehensive AI approaches to the treatment plan. We tried to answer this question by adding the following sentences:

On the “Comprehensive Perspective for the Next Generation of CADs” section:

“There are three main levels of actions for the CADs: screening, diagnosis/characterization, and treatment assignment. On the treatment planning, the novel CADs would assist the clinicians in choosing the optimal treatment, with clear and transparent explanations for the recommendations. Currently, recommendations are clear about the need for biopsy to determine mutational status of the tumor in order to select the best available treatment [1]. However, learning from a comprehensive approach, offers the possibility of capturing correspondences between processes and gaining an in-depth understanding of pathophysiological changes, which allow to select of a personalized treatment that will improve effectiveness and efficiency, while diminishing avoidable therapy-related adverse events. The AI-based approaches would stratify the patient groups according to their singular properties, in order to more effectively evaluate the potential efficacy of treatments, recommend the sequence of therapies, and predict the effects of specific drugs and clinical outcomes. This strategy may be particularly helpful in elderly or unfit patients who are at higher risk of procedure-related complications. In these patients, these risks may unable a proper diagnosis or treatment. For example, since inoperable early stage lung cancer patients may benefit of stereotactic radiotherapy [2], an non-invasive diagnosis strategy in this group of patients could diminish the risks, prevent complications associated to the traditional methods and allow prompt treatment leading to a better disease control and survival.  Moreover, CADs would use aggregated knowledge of many patients with matching results, history, biomarkers, physiological characteristics, and behavioral risk to present the clinicians with the most efficacious treatment option. In addition to prevent toxicities, the personalized treatment recommendations can aid to reduce time loss and costs.”

On the Introduction Line 49:

“Despite the evident benefits, with the increased use of these personalized therapies in oncology, new side effects have emerged causing important clinical challenges in the management of lung cancer patients. In fact, although the majority of these events are mild, some of them can be severe and potential life-threatening [3].”

Introduction Line 87:

…help therapeutic decision and leverage the personalised medicine, selecting patients who will definitely benefit of target therapies and avoid superfluous therapy-related side effects.

References

[1]      N. I. Lindeman et al., “Updated molecular testing guideline for the selection of lung cancer patients for treatment with targeted tyrosine kinase inhibitors guideline from the college of American pathologists, the international association for the study of lung cancer, and the association for molecular pathology,” in Archives of Pathology and Laboratory Medicine, 2018.

[2]      G. M. M. Videtic et al., “Stereotactic body radiation therapy for early-stage non-small cell lung cancer: Executive Summary of an ASTRO Evidence-Based Guideline,” Pract. Radiat. Oncol., 2017.

[3]      J. B. A. G. Haanen et al., “Management of toxicities from immunotherapy: ESMO Clinical Practice Guidelines for diagnosis, treatment and follow-up,” Ann. Oncol., 2017.

Round 2

Reviewer 1 Report

In this revision of the article, the authors have addressed my previous comments by adding clarifying sentences in the article and including additional references. I only have a few minor comments for this version of the article.

The reference for the lines 58-59, "CT is more sensitive than chest radiography in lung cancer-screening", would probably be better if some of the original papers were cited rather than a review, such as: Results of Initial Low-Dose Computed Tomographic Screening for Lung Cancer | NEJM. It is also worth citing the results of the NELSON trial (although that trial did not compare to x-ray): Reduced Lung-Cancer Mortality with Volume CT Screening in a Randomized Trial | NEJM.

There are a few awkward sentences that still remain. Lines 76-77 I pointed out in the previous review, but it could be reworded to be "…solutions enable the identification of… that are not visible to…". Lines 86, it should be "benefit from". Line 91, should just be "…have identified…". Line 137, "…which allow the capture of information that is not visible to the naked eye and avoids…". Line 161, "…which allow the selection of…". Line 168, "…benefit from stereotatic…" Line 173, "In addition to preventing toxicities…".

Author Response

Comments and Suggestions for Authors

In this revision of the article, the authors have addressed my previous comments by adding clarifying sentences in the article and including additional references. I only have a few minor comments for this version of the article.

Author response: We would like to thank the reviewers for providing constructive comments that helped improve the manuscript.

Reviewer#1, Concern # 1: The reference for the lines 58-59, "CT is more sensitive than chest radiography in lung cancer-screening", would probably be better if some of the original papers were cited rather than a review, such as: Results of Initial Low-Dose Computed Tomographic Screening for Lung Cancer | NEJM. It is also worth citing the results of the NELSON trial (although that trial did not compare to x-ray): Reduced Lung-Cancer Mortality with Volume CT Screening in a Randomized Trial | NEJM.

Author response: The authors thank the reviewer for this relevant insight. We completely agree about the relevance of the references, and all of them were added to the manuscript.

Reviewer#1, Concern # 2: There are a few awkward sentences that still remain.

Lines 76-77 I pointed out in the previous review, but it could be reworded to be "…solutions enable the identification of… that are not visible to…".

Lines 86, it should be "benefit from".

Line 91, should just be "…have identified…".

Line 137, "…which allow the capture of information that is not visible to the naked eye and avoids…".

Line 161, "…which allow the selection of…".

Line 168, "…benefit from stereotatic…"

Line 173, "In addition to preventing toxicities…".

Author response: We thank the reviewer to identify those mistakes. All of them were changed.
